# Plant-Derived Waste as a Component of Growing Media: Manifestations, Assessments, and Sources of Their Phytotoxicity

**DOI:** 10.3390/plants13142000

**Published:** 2024-07-22

**Authors:** Juncheng Liu, Wenzhong Cui, Zhiyong Qi, Lingyi Wu, Wanlai Zhou

**Affiliations:** 1School of Mechanical Engineering, Chengdu University, Chengdu 610106, China; liujuncheng@stu.cdu.edu.cn; 2Institute of Urban Agriculture, Chinese Academy of Agricultural Sciences, Beijing 100875, China; cuiwenzhong@stu.cdu.edu.cn (W.C.); qizhiyong@caas.cn (Z.Q.); 82101225393@caas.cn (L.W.)

**Keywords:** plant-derived waste, phytotoxicity, growing media, phenolic compounds, organic acids

## Abstract

Every year, approximately 2 billion tons of plant-derived waste (such as straw and crop residues) are generated globally, most of which are either incinerated, dumped, or landfilled without proper planning, leading to severe environmental pollution and resource wastage. Plant-derived waste exhibits potential advantages as a growing media component in various aspects. However, numerous studies have also indicated that plant-derived waste generally possesses strong phytotoxicity, which must be removed or reduced before being utilized as a growing media component. Therefore, accurately assessing their phytotoxicity and appropriately modifying it to ensure their support for plant growth when used as a growing media component is crucial. This paper reviews the manifestation and assessment methods of phytotoxicity in plant-derived waste; systematically summarizes the phytotoxicity sources of three common types of plant-derived waste (garden waste, crop straw, and spent mushroom substrate), as well as the toxic mechanisms of two representative phytotoxic substances (phenolic compounds and organic acids); and proposes some insights into further research directions. By consolidating insights from these studies, this review aims to deepen our understanding of phytotoxicity and its implications, and offer valuable references and guidance for future research endeavors and practical applications.

## 1. Introduction

Plant-derived waste encompasses a wide range of materials derived primarily from plants, such as fruit and vegetable refuse, urban garden waste, straw from various crops, and waste from forestry and agricultural processing. The primary constituents of an edible mushroom cultivation medium consist of wood fragments, crop residues, and similar materials. As a result, edible mushroom residues are classified as plant-derived waste for the purposes of this article. Agricultural by-products (straw and rice straw) are generated on a worldwide scale at an estimated annual rate of 2 billion tons [1]. Additionally, a substantial quantity of forestry refuse and other waste derived from plants is present. At present, a significant proportion of plant-derived waste is disposed of in unplanned landfills, discarded, or incinerated, leading to detrimental environmental impacts and resource wastage [2,3,4].

Growing media are essential inputs in soilless cultivation. Global cultivation substrate consumption reached 67 million m^3^ in 2017 and is projected to surge to 283 million m^3^ by 2050 [5]. Peat has been the most extensively utilized horticultural substrate worldwide for the past several decades on account of its cost-effectiveness and superior physical, chemical, and biological properties [5,6]. An anticipated annual global production of peat amounts to approximately 90 million m^3^, of which 40 million m^3^ is utilized for horticultural purposes [7]. In total, 30% of the total global soil carbon and 75% of the total atmospheric carbon are stored in peatlands, making them the largest and most significant natural carbon reservoir on land [8]. Nevertheless, peatland ecosystems are exceedingly delicate, and peat mining not only results in the depletion of peatland biodiversity but also contributes to the global carbon cycle imbalance by releasing carbon back into the atmosphere [9]. Consequently, the horticultural sector has been compelled to curtail or diminish its reliance on peat [10], thereby emphasizing the criticality of sustainable alternatives to peat.

Plant-derived waste has numerous potential advantages as growing media components for cultivation. To begin with, plant-derived waste is predominantly composed of plant fibers, which possess a multitude of vital essential nutrients for plants (e.g., N, P, and K) and can be readily converted into substrate materials that exhibit favorable water retention and air permeability. Additionally, the vast quantities and broad distribution of plant-derived waste allow for its local supply, a feature that numerous scholars consider to be critical [11,12]. Furthermore, the process of converting plant-derived waste into growing media components exemplifies upcycling, as it transforms waste materials into valuable resources. This reduces the dependence on peat as a growing medium, thereby decreasing peat extraction and lowering global carbon emissions.

Due to the numerous benefits associated with plant-derived wastes, researchers from across the globe have devoted significant efforts in recent decades to examining diverse plant-derived wastes as potential alternatives to peat [13,14,15,16,17,18]. The aforementioned studies have provided evidence for the viability of utilizing plant-derived waste as a constituent of cultivation growing media. However, they have also unveiled the primary challenge that plant-derived waste encounters when utilized as such: phytotoxicity [19]. Delay in seed germination, the inhibition of plant growth, or any other detrimental consequence inflicted upon the plant as a result of a particular substance (phytotoxin) or growth conditions constitute phytotoxicity [20,21,22,23]. For instance, Chemetova et al. [12] found that phenolic compounds in *Acacia melanoxylon* bark inhibit germination and root elongation in Lepidium sativum. In a separate study, Wanlai Zhou et al. [23] observed that untreated green waste reduces the germination rate of *Brassica* rapa chinensis and detected significant levels of malic acid and succinic acid within the green waste. Furthermore, Inês A. Pinho et al. reported in a research article [24] that olive residue decreases root length and germination rates in Lepidium sativum, potentially leading to nutrient deficiencies in most *Lepidium sativum* specimens.

The subject of phytotoxicity has garnered more attention in recent years as an expanding number of studies have sought alternatives to peat. Regrettably, the current state of knowledge does not include a comprehensive analysis of the phytotoxicity exhibited by the residues derived from plants. This paper presents an examination of the various forms of phytotoxicity, the methods of assessment, the sources of phytotoxicity in three prevalent plant-derived wastes (garden wastes, crop residues, and fungal residues), and the toxicity mechanisms of two representative phytotoxic substances (phenolics and organic acids), and offers some suggestions for future research in this area of study. The systematic review of these studies is anticipated to contribute to the body of knowledge on phytotoxicity and serve as a resource and guide for future research and applications in this field.

## 2. Manifestations of Phytotoxicity

### 2.1. Phytotoxicity Caused by Plant-Derived Waste

Phytotoxicity is characterized by its capacity to obstruct seed germination and plant growth [25]. Specifically, germination inhibition manifests as root non-germination, leaf yellowing and browning, root curling, and root shortening, as illustrated in Figure 1. Conversely, phytotoxicity in the context of ensuring normal nutrient and water availability growth inhibition typically presents as leaf discoloration and stunted plant stature, as depicted in Figure 2. This is consistent with the findings of several previous researchers [22,23]. The particular phytotoxic responses vary significantly among plant species due to differences in tolerance levels. Additionally, the phytotoxic effects prompted by various toxigenic sources may also differ.

The phytotoxicity of plant-derived detritus is often linked to the presence of volatile or water-soluble phytotoxic compounds [25]. Common phytotoxic organic chemicals include aromatic hydrocarbons, aldehydes, ketones, carboxylic acids, methoxyphenols, and catechols [25,26]. The manifestation of phytotoxicity is contingent upon the form and concentration of the source substance. Notably, the same homotoxic substance can exhibit varying phytotoxic effects depending on its state and concentration levels. For instance, Lynch et al. [27,28] observed that low concentrations of acetic acid and malonic acid enhance seedling root elongation, while higher concentrations of organic acids impede plant growth. Similarly, Gajalakshmi et al. [29] reported that the toxicity of chromium (Cr) to plants is contingent upon its ionic form, with trivalent chromium (Cr^3+^) being significantly less toxic than hexavalent chromium (Cr^6+^). The current literature on phytotoxicity predominantly addresses seed germination and seedling growth, with limited exploration into the physiological impacts of substrate materials on plants, such as photosynthesis, nutrient uptake, and stress resistance.

### 2.2. Phytotoxicity and Allelopathy

Although phytotoxicity and allelopathy are interrelated, they represent fundamentally distinct phenomena. Allelopathy refers to both the detrimental and beneficial effects that secondary metabolites from plants, bacteria, fungi, and algae can exert on the growth and development of other organisms within natural ecosystems and agricultural environments [30]. In natural settings, plants release allelochemicals to secure a competitive advantage. Intriguingly, while these chemicals may confer benefits to the producing plant, they can adversely affect neighboring flora [31]. Many allelochemicals, such as phenolic acids, organic acids, and terpenoids, are secondary metabolites produced via the primary metabolic pathways involving carbohydrates, lipids, and amino acids [32]. Table 1 presents a comparison of allelopathy and phytotoxicity from various perspectives.

## 3. Methods of Phytotoxicity Assessment

### 3.1. Chromatographic and Spectroscopic Techniques

Phytotoxicity is primarily attributed to specific phytotoxic compounds; therefore, the accurate identification of these compounds within substrate materials is essential (Table 2). Hitzl et al. [33] developed a gas chromatography technique that facilitated a preliminary, yet rapid, assessment of the phytotoxic potential of hydrothermal carbon. The results demonstrated a statistically significant positive correlation between the concentration of the volatile compounds detected in the material through gas chromatography and its phytotoxic effects, as determined by seed germination assays. This correlation substantiates the effectiveness of the analytical approach. In a related study, Zhou et al. [23] employed gas chromatography to investigate the composition of organic matter in six distinct types of plant-derived waste. Their investigations revealed that the samples exhibiting increased concentrations of succinic acid, malic acid, citric acid, and quinic acid (1.0 mg/mL) were consistently correlated with heightened phytotoxicity levels; which agreed with previous results [28]. This observation lends credence to the hypothesis that compositional analysis can be an effective tool for evaluating phytotoxic potential. In a complementary study, Cui et al. [34] utilized fluorescence spectroscopy to determine the excitation–emission matrix (EEM) spectra of a diverse array of organic waste materials, including chicken manure, swine manure, food waste, weeds, straw, leaves, and a variety of fruits and vegetables. They further employed projection pursuit regression to develop a methodology that facilitates the rapid and accurate determination of phytotoxicity levels during the composting process. This advancement provides composting facilities with a valuable tool for the effective monitoring of compost maturity.

### 3.2. Bioassay

The vast array of phytotoxic compounds inherent in plant materials presents a formidable challenge to their exhaustive analysis and identification, rendering such an endeavor nearly insurmountable. Furthermore, a significant proportion of persistent organic pollutants (POPs) exhibit physiological activity, thereby exerting an influence on plant growth and development that is comparable to synthetic hormones [35]. Notably, these compounds can elicit a biological response at exceedingly minute concentrations, often eluding detection due to their presence below the established analytical thresholds [35]. As a result, evaluations based on biological responses may be deemed more effective and immediate. These assessments employ plant seeds and seedlings within bioassays, aligning with the established protocols such as the seed germination test (ISO 11269-1 [36], CEN 16086-2 [37]) and seedling growth test (ISO 18763 [38], CEN 16086-1 [39], OECD TG 208 [40]).

#### 3.2.1. Seed Germination Experiment

Different crops demonstrate a spectrum of sensitivities to various phytotoxic agents, and the selection of indicator plants for seed germination assays does not adhere to a uniform standard. Commonly, species such as watercress, cabbage, and lettuce are employed in these assays. Typically performed in Petri dishes, seed germination tests are categorized into direct contact, indirect contact, or water-soaked extract methods, contingent upon the mode of interaction between the test substance and the seed (Figure 3).

The direct contact method effectively encompasses the effects of the myriad substances contained within the material, encompassing both hydrophilic and hydrophobic constituents. Consequently, the results derived from this method are theoretically more representative of the actual impact exerted by the substrate material upon implantation. Ortegaet et al. [41] conducted a comparative analysis of the aqueous leachate method and the direct contact method, deducing that the latter offers a more accurate reflection of phytotoxicity in growing media, attributable to its uncomplicated testing process and the greater correlation between the experimental outcomes and the genuine growth conditions. Nonetheless, the capacity of a research methodology to discern nuanced differences among various treatments and compounds is of paramount importance. This becomes particularly challenging with the direct contact method, as it tends to amplify the inhibitory effects on plant growth more than the indirect contact and aqueous leachate methods. Consequently, employing water leachate and indirect contact techniques may be advantageous when the physical characteristics of the material being analyzed exert a significant influence on plant development (Table 2).

Fujii et al. [42] developed the ‘sandwich method’, an indirect contact technique utilized in seed germination assays. In this protocol, a measured amount of test material, ranging from 10 to 50 mg, is placed on the lower surface of a Petri dish. Subsequently, two layers of agar medium are introduced to encapsulate the material; seeds are then sown atop the upper agar layer. Through this stratified medium, the phytotoxic properties of the test substance are indirectly imparted to the seeds during germination testing. The sandwich method’s design, which precludes direct seed–material contact, typically results in higher germination indices compared to those observed with direct contact methods.

Seed germination assays employing aqueous extracts are favored for their simplicity over the more intricate sandwich method, thus becoming the predominant technique for phytotoxicity evaluation. Typically, a defined ratio (commonly 1:10) of the test material is water-extracted, and the resultant solution is used to saturate germination paper for the assay. This approach aligns with the established phytotoxicity assessment standards such as ISO 11269-1 and CEN 16086-2. The International Organization for Standardization (ISO) developed ISO 11269-1, outlining a protocol to determine the adverse effects of contaminants in growing media or soils on plant root systems. This standard is relevant for assessing a range of substances including chemicals, composts, and soils. Similarly, the European Committee for Standardization (CEN) introduced CEN 16086-2, which assesses the initial impact of culture media and soil amendments on plant germination and root development. While aqueous extract-based germination tests shed light on the influence of water-soluble phytotoxic agents, gauging the effects of the volatile compounds within materials remains a significant analytical challenge [43].

#### 3.2.2. Seedling Growth Experiment

Seedling growth assays involve the examination of seedling development within a composite substrate, integrating the test material with a standard growth medium, such as peat, in various ratios. The International Organization for Standardization (ISO) established ISO 18763, a standard commonly utilized to assess the effects of solid or liquid chemical toxicants on plant growth within soils or substrates contaminated by such agents (e.g., compost and sewage).

The European Committee for Standardization (CEN) established CEN 16086-1, a standard that is regularly utilized to assess the adverse effects of traditional soil amendments and growing media on the growth of *Cabbage*. Concurrently, the Organisation for Economic Co-operation and Development (OECD) promulgated OECD TG 208, a harmonized experimental protocol designed to evaluate the environmental repercussions of various substances by monitoring the early stages of plant development.

CEN 16086-1 delineates the recommended ratios of various materials for testing. The prepared samples are then sown with test plants and cultivated under controlled conditions, ensuring optimal temperature, light intensity, and humidity. It is imperative to provide adequate hydration and nutrients throughout the testing period. Phytotoxicity is indicated by the germination and subsequent growth of the test plants within a predetermined duration. Specifically, CEN 16086-1 mandates a five-day germination window for *Cabbage*. Following the emergence of at least fifty percent of the plants with five or more leaves, an immediate evaluation of plant biomass is required.

Contrary to the methodology prescribed by CEN 16086-1, the assessment of phytotoxicity according to OECD TG 208 is customarily conducted within a period ranging from 14 to 21 days following the attainment of 50% germination in the control group. This evaluation employs an exhaustive testing protocol that encompasses a multitude of parameters, including but not limited to, the height of the seedlings, their dry and fresh weights, the germination rate, and any observable adverse effects on the various morphological components of the plants. The primary objective of the seedling growth assay is to ascertain the prospective influences of chemical agents on both the germination phase of plant seeds and the subsequent initial stages of seedling development. This assay is applicable for evaluating the toxicological profiles of a broad spectrum of substances, including general chemicals, pesticides, and biocides. Nonetheless, its applicability does not extend to gaseous chemical entities. Moreover, it does not encompass the assessment of protracted effects or the influence on reproductive cycles, such as the processes of fruit maturation, blossoming, or the formation of seeds.

Seed germination assays offer the advantage of swiftly determining the influence of potential phytotoxic agents on plant ontogeny by gauging the performance of seed germination. Nonetheless, a notable limitation of these assays is their focus on the incipient phase of plant growth, which may not provide a faithful representation of the plant’s developmental trajectory under field conditions. Generally, germination assays necessitate a shorter duration for execution compared to seedling growth assays. While a discernible correlation between seed germination and seedling growth assays is typically observed, it is imperative to acknowledge that this relationship is not invariable. It is contingent upon the specific properties of the phytotoxic agent employed in the seedling growth assay and the prevailing substrate conditions. For instance, the amelioration and dilution effects of the substrate may result in a markedly enhanced performance of the plant within a controlled environment relative to outcomes observed in seed germination assays [26].

## 4. Sources and Composition of Phytotoxicity

### 4.1. Phytotoxicity of Representative Plant-Derived Wastes and Their Sources

#### 4.1.1. Garden Waste

Refuse derived from urban greening activities, such as leaves, lawn clippings, and tree residues, is commonly designated as ‘garden waste’. This category of garden waste is predominantly composed of lignin, hemicellulose, and cellulose, and is rich in essential nutrients, namely nitrogen, phosphorus, and potassium. These components exhibit a considerable potential for substrate utilization. It is projected that the annual production of horticultural waste in China will attain an approximate volume of 310,400 metric tons [44].

Numerous studies have confirmed the feasibility of integrating garden waste into cultivation substrates while also recognizing that, on average, garden waste exhibits significant phytotoxicity [2,22,45]. Despite the emergent nature of research on plant residue phytotoxicity, part of phenolic compounds and organic acids have been identified as principal phytotoxic agents (Table 3). Extensive prior research has demonstrated a direct correlation between the phenolic content in plant residues and their inhibitory impact on seed germination [46,47], Furthermore, Chemetova et al. have indicated that phenolic compounds are the primary contributors to phytotoxicity [20,25]. Tetrahydroxy styrene, along with its glucoside epigallocatechin and proanthocyanidins, have been identified as the agents responsible for the phytotoxicity of *Sargasso pine* bark [48]. Additionally, the phytotoxic effects observed in debris from *Giant birch* and *Red gum trees* have been linked to naringenoids [49]. Politycka et al. [46] attributed the phytotoxicity of sawdust and pine bark to seven specific phenolic acids. Similarly, the growth of plants was hindered by the aqueous extracts of bark rich in phenolic compounds [50]. Bark and sawdust preparations from *Japanese red cedar* and *Willow were* found to contain phenolic acids and tannins that significantly inhibit the growth of trifoliate orange and rice seedlings [51]. Machrafi et al. [52] identified a spectrum of fourteen phenolic compounds in the residues of *White pine* bark aged from fresh to twenty years. It was also noted that the seed germination index negatively correlated with both the total phenolic content and specific phenolic compounds. Contrarily, Zhou et al. [23] reported that the phytotoxicity of six representative garden wastes did not significantly correlate with the total phenol content. However, strong positive correlations were found with organic acid and amino acid content, suggesting that these substances, and their derivatives, are likely the primary contributors to the phytotoxicity observed in these garden wastes.

#### 4.1.2. Crop Stalks

Post-harvest, crop straw constitutes the residual by-products left in the field, including rice straw, rape straw, wheat straw, and maize stalks. Recognized as a valuable biomass resource, straw finds diverse applications in feed, energy, and fertilizer, and as industrial raw materials [62]. Given China’s status as a major agricultural nation, its straw output is significant, with peak production of 8.02 × 10^8^ tons recorded in 2021 (National Bureau of Statistics). Globally, straw production can reach an estimated 1.5 × 10^9^ tons annually [63]. Rich in trace elements such as nitrogen, phosphorus, and potassium, straw promotes the growth and development of crops [64]. Compared to garden waste, crop residues generally have lower levels of phenolics and organic acids, suggesting a higher potential for substrate utilization.

Straw substrate utilization is not without its phytotoxicity concerns, as unstable agricultural residues may induce phytotoxic effects, a phenomenon partially attributed to substances generated during decomposition [65]. For example, under anoxic conditions, wheat straw residues can convert cellulose to acetic acid, potentially reducing crop yields [66]. A soil amendment with 5% tomato pomace and 2% mature compost has been shown to cause soil acidification and impede lettuce emergence [67]. While straw enhances soil fertility and nutrient recycling, it can also negatively affect crops. Decomposition by-products such as phenylglycolic acid, p-cyanobenzoic acid, vanillic acid, and acetic acid may inhibit the growth of rice seedlings [68,69]. Similarly, organic compounds from wheat straw decomposition can adversely affect various plant species [70]. In contrast, garden waste contains higher levels of soluble organic matter and hemicellulose, which are susceptible to rapid microbial degradation in soilless cultures, leading to oxygen depletion and the subsequent inhibition of plant growth [66].

#### 4.1.3. Spent Mushroom Growing Media

Mycorrhizae, commonly referred to as spent mushroom substrate (SMS), constitute the organic residue remaining after mushroom harvest. The production process of edible mushrooms is such that for every kilogram harvested, approximately five kilograms of fresh mushroom waste is generated [71]. In 2020, China reported a production of edible mushrooms totaling 40,614,200 metric tons according to the China Edible Fungi Association (http://bigdata.cefa.org.cn/). This implies that the annual yield of mushroom by-products in China exceeded 200 million metric tons [72].

Mycorrhizal residues, primarily composed of decomposed biomass such as wood chips, maize cobs, cottonseed hulls, animal manure, straw, and high concentrations of soluble organic matter like organic acids and amino acids, also include residual additives such as gypsum, lime, and various nutrients [71]. These components enrich the nutrient profile of the residues and confer excellent water retention properties, making them suitable for use as a substrate. Given that the cultivation of edible mushrooms and horticultural activities often occur in suburban areas, the application of mushroom waste in horticulture presents a considerable opportunity. Research has demonstrated the feasibility of substituting mycorrhizal residues for peat in horticultural mediums [73,74,75].

Numerous agricultural studies have reported reductions in crop yields when SMS is incorporated into substrates. Utilizing over 50% SMS in the substrate adversely affects the growth of *Ginseng* and *Cotton grass* [21]. Similarly, an excess of mushroom residue in growing media impedes *Watercress* germination and root elongation [76]. Medina et al. [77] examined the effects of blending fresh *Agaricus bisporus* and *Agaricus flatus* mushroom pomace with peat in proportions ranging from 25% to 75% on the growth of *Tomato*, *Eggplant*, and *Chayote*. The findings indicated that higher pomace ratios led to increased growth inhibition in all the tested crops. Analogous results were observed with chili peppers cultivated using fresh *Almond mushroom* pomace [78]. Such declines in biological yield are commonly attributed to the phytotoxic properties of freshly harvested mushroom residues.

Numerous researchers have documented that mushroom residues exhibit elevated electrical conductivity (EC), which, in turn, can adversely impact plant growth and development. Szmidt et al. [79] reported the EC values of mushroom residues ranging from 1.4 mS·cm^−1^ to approximately 2.4 mS·cm^−1^. Typically, fungal residues exhibit high EC values, often surpassing the recommended range of 0.5–1.5 ms·cm^−1^ [80], which can impede crop growth and development. This assertion is supported by various studies; for example, a growth medium with SMR containing higher EC values (2.3 dSm^−1^) negatively affected plant growth [78]. The phytotoxicity of SMS is also linked to high concentrations of phytotoxic chemicals such as aromatic compounds, phenolic compounds from lignocellulose, and organic acids. Chen et al. [72] found that superheated steam roasting treatment significantly reduced the levels of organic acids and phenolics in shiitake mushroom pomace, leading to a marked improvement in the seed germination index.

### 4.2. Common Phytotoxic Substances in Plant-Derived Waste

#### 4.2.1. Phenolic

The primary phytotoxic substances present in plant-derived residues are phenolics, organic acids, and their derivatives [23]. Phenolic compounds, including monophenols, phenolic acids, tannins, and flavonoids, are biosynthesized in plants through metabolic pathways involving mangiferolic acid and acetic acid [81]. While these compounds are endogenously produced by plants as a defense against diseases and infections, their accumulation in the environment can lead to intoxication and hinder plant development and growth. Notably, a majority of phenolic compounds are recognized for their allelopathic properties [81].

Phenolic compounds exert significant biological effects on plants, which include the following: (1) Increased cell membrane permeability, leading to the leakage of cellular contents and enhanced lipid peroxidation, potentially causing slow growth or plant death [81]. (2) The impediment of nutrient absorption, disrupting plant development. (3) The inhibition of root growth and cell division, affecting overall plant growth [81]. (4) Decreased oxygen uptake capacity, along with reduced chlorophyll content and photosynthetic rate [82]. (5) The reduction or deactivation of phytohormone activity, hindering normal physiological processes [83]. (6) Interference with protein synthesis, particularly by phenolics like ferulic and cinnamic acids [84].

The phytotoxic effects of phenolic compounds are closely associated with their molecular structures. Research indicates that lipophilic substances can easily cross cellular membranes and are inherently phytotoxic [60]. Furthermore, the extent of phytotoxicity is strongly linked to the number of lipophilic structures present in these compounds. A study by Pinho et al. [24] on eleven phenolic compounds revealed that an increase in the number of -OH and -OCH_3_ groups reduced both the lipophilicity and the phytotoxic effects on watercress. Conversely, cinnamic acid, which consists solely of lipophilic carbon chains, demonstrated the most significant phytotoxicity. The study also found that the phytotoxic effects of various phenolic mixtures were neither additive nor synergistic; instead, the overall phytotoxicity was determined by the compound with the highest lipophilicity.

The concentration of phenolic compounds is a critical factor influencing their phytotoxic effects. Lignans isolated from brassica napus were found to inhibit the growth of *Lettuce seedlings* at very low concentrations (1 nmol/L), as reported by Cutillo et al. [85]. Gallic acid, in concentrations of 2–4 mg/L, stimulates the growth of *Watercress* roots but becomes phytotoxic at higher concentrations [24]. Chou and Leu [86] analyzed the phytotoxic effects of various phenolic acids, including protocatechuic acid, chlorogenic acid, gallic acid, p-hydroxybenzoic acid, caffeic acid, 3,5-nitrobenzoic acid, and 3,4-dihydroxybenzaldehyde, on *Lettuce* seeds at concentrations ranging from 10 to 500 mg/L. Their findings revealed that the inhibitory effects exceeded 30% at a concentration of 10 mg/L, with the phytotoxicity increasing as the concentration of the phenolic compounds increased.

The EC_50_, or median effective concentration, is commonly used to evaluate the toxicity of environmental chemicals towards organisms, reflecting the concentration at which a compound exerts half of its maximal effect. For sensitive species like rye and wheat, the EC_50_ values for phenolic acids, such as p-coumaric acid and p-hydroxybenzoic acid, ranged from 1 mg/L to 10 mg/L. In contrast, more tolerant species like *Cabbage* exhibited higher EC_50_ values for compounds like phenol and o-cresol, measured at 125.6 mg/L and 54.9 mg/L, respectively [53]. *Watercress* demonstrated the greatest sensitivity to cinnamic acid with an EC_50_ of 60 mg/L, followed by phenol with 100 mg/L. Other phenolic acids, including vanillic acid, p-coumaric acid, protocatechuic acid, and veratric acid, had EC_50_ values of 180 mg/L, 190 mg/L, 400 mg/L, and 500 mg/L, respectively [24].

#### 4.2.2. Organic Acids

Organic acids significantly influence plant seed germination, as they can alter the permeability of the seed cell membranes, affect the activity of crucial metabolic enzymes, and impede root development during the second phase of germination, ultimately reducing germination rates and limiting root growth [87,88,89,90]. Cocucci et al. [91] investigated the effects of butyric acid on the germination of *cosmos seeds* (Phacelia tanacetifolia Benth. cv. Bleu Clair), finding that it obstructed the initial stages of seed germination, particularly under dark conditions. Butyric acid was also observed to inhibit various metabolic activities, including respiratory activity; levels of reducing sugars, glucose-6-phosphate, and malate; CO_2_ fixation in the dark; transport activity; and macromolecular synthesis. Additionally, the acidic conditions created by organic acids can negatively impact plant development [53].

The molecular structure of organic acids is pivotal in determining their phytotoxicity. For example, *Barley* root elongation was minimally affected by acetic acid, citric acid, lactic acid, and glycine, while other aliphatic and aromatic organic acids, and amino acids showed inhibitory effects at concentrations as low as 5 mM [27]. *Chard* seeds exhibited greater germination inhibition when exposed to malic acid compared to citric acid at the same concentration [88]. Generally, the phytotoxicity of lower fatty acids increases with the length of the carbon chain, except for formic acid [27].

Furthermore, the concentration of organic acids correlates positively with their phytotoxic effects. In conditions with a pH of 6.5, the low concentrations of acetic and malonic acids enhanced the growth of seedling roots, while higher concentrations above 1.0 mg/mL significantly hindered seed germination [23,27,89].

## 5. Conclusions and Prospect

This paper reviews the manifestations of phytotoxicity in plant-derived waste and systematically summarizes the phytotoxicity sources of three common types of plant-derived waste as well as the toxic mechanisms of phenolic compounds and organic acids. This study provides a detailed summary of the methods for assessing the phytotoxicity of plant-derived waste, analyzing their applicability, advantages, and disadvantages. However, there is still a lack of mature phytotoxicity assessment standards in this field.

Currently, research on the phytotoxicity of plant-derived waste is relatively fragmented globally. In the field of the substrate utilization of plant-derived waste, we propose the following future research directions: The development of evaluation standards: establish evaluation standards for the phytotoxicity of plant-derived waste to ensure precise assessments. The study of physicochemical properties: Investigate the physicochemical, biological, and environmental properties of plant-derived waste substrates. Evaluate their effects on plant growth, stress resistance, disease resistance, quality, and yield, and optimize application methods and management practices. Toxicity reduction techniques: develop techniques to reduce the phytotoxicity of plant-derived waste, such as the addition of bio-agents, slow-release fertilizers, and biochar, to mitigate the negative impacts of phenolic compounds or organic acids on plants. Composite utilization techniques: Explore the composite utilization techniques of plant-derived waste substrates with other organic or inorganic materials, such as agricultural waste, industrial waste, and slag. This aims to create multifunctional composite cultivation substrates, thereby expanding their application fields and scope.

## Figures and Tables

**Figure 1 plants-13-02000-f001:**
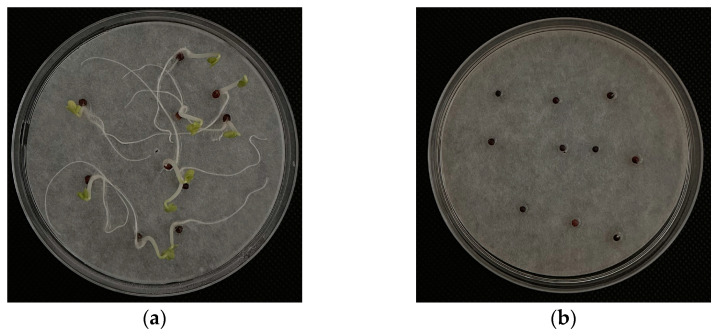
Photographs of garden waste germination experiment ((**a**) is deionized water and (**b**) is garden waste extract).

**Figure 2 plants-13-02000-f002:**
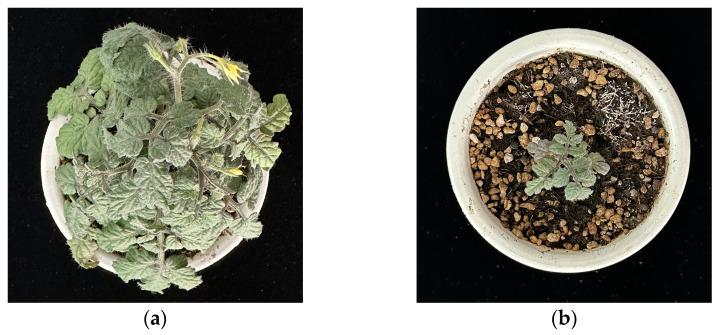
Photographs of experiments with potted *tomato* plants from garden waste ((**a**), peat control; (**b**), 50 percent peat–garden waste mix).

**Figure 3 plants-13-02000-f003:**
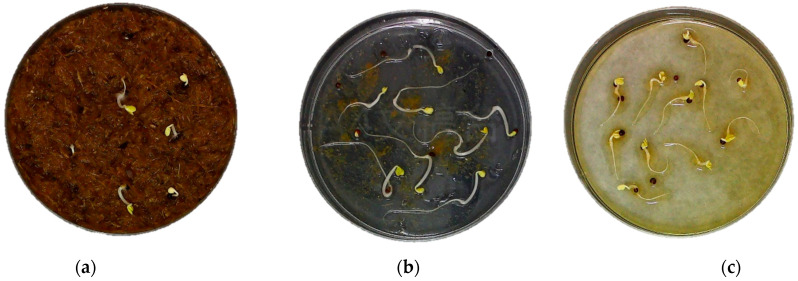
Photographs of seed germination tests by the direct contact, sandwich, and aqueous extract methods ((**a**) is the direct contact method, (**b**) is the sandwich method, and (**c**) is the aqueous extract method).

**Table 1 plants-13-02000-t001:** Difference between phytotoxicity and allelopathy.

	Source	Target	Effect
Phytotoxicity	Chemical compounds derived from the environment or adverse environmental conditions	Plants	Harmful
Allelopathy	The secondary metabolites of plants, bacteria, fungi, and algae	Organisms in agricultural and natural ecosystems	Harmful and beneficial

**Table 2 plants-13-02000-t002:** Methods of phytotoxicity assessment.

Methods		Merit	Flaw
Chromatographic and spectroscopic techniques		Capable of detecting a diverse array of compounds	High cost and the effects of compounds on plants need to be verified by further cultivation
Seed germination experiment	direct contact methods	More precisely reflects the impact of substrate materials during cultivation	May exacerbate adverse effects on plants
	indirect contact methods	Avoid direct seed contact with the substrate material	Subtle or latent adverse effects may not be readily observable
	water-soaked extract methods	A stronger correlation exists among actual growth conditions	Volatile compounds cannot be reliably quantified using this method
Seedling growth experiment	ISO 18763	Enhanced accuracy in assessing early plant growth outcomes	Limited scope and complex implementation
	CEN 16086-1	The improved assessment of plant viability	The only choice for plants is *Cabbage*
	OECD TG 208	Plants can be assessed based on a range of physiological indicators	Persistent phytotoxic effects remain challenging to assess

**Table 3 plants-13-02000-t003:** Negative effects of some phenolic compounds and organic acids on plants.

	Concentration	Plants	Phytotoxicity Manifestations
Carbolic acid [53]	125.6 (mg/L)	*Brassica rapa chinensis*	Inhibits root elongation.
ortho-Cresol [53]	54.9 (mg/L)	*Brassica rapa chinensis*	Inhibits root elongation.
Secolignan [54]	1 (nmol/L)	*lettuce*	Seed germination was still significantly inhibited at very low concentrations.
PAHS [55]	100 (mg/L)	*Lepidium sativum*	Inhibits seed development and root elongation.
Ethanol [56]	2500 (mg/L)	*Euphorbia heterophylla*	Delayed seed germination and growth inhibition.
Acetic acid [27]	300 (mg/L)	*Oryza sativa*	Root growth inhibition up to 25 percent.
Gallic acid [57]		*lettuce*	The inhibition of lettuce growth and development.
Ferulic acid [58]		*lettuce seeds*	Inhibits the germination of lettuce seeds.
Acetic acid, propionic acid, butyric acid [59]	60.05 (mg/L)	*Oryza sativa*	Seedlings wilting and dehydration-like symptoms.
Acetic acid, propionic acid, butyric acid [59]	300.25 (mg/L)	*Oryza sativa*	The inhibition of root growth in seedlings and bronze-like symptoms in leaf tips.
Acetic acid, propionic acid, butyric acid [59]	600.5 (mg/L)	*Oryza sativa*	Reduced plant height and seedling death within 24 h.
Cinnamic acid [60]	35 (mg/L)	*Phaseolus vulgaris*	Affects seedling development, including seedling root length, germination, and fresh weight. Growth and concentration showed a negative correlation.
Caffeic acid [61]		*Vigna radiata*	Influence on the early growth and morphology of mung bean hypocotyl plugs.

## Data Availability

All relevant data are within the paper.

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
