# Peer review of "Plant-Derived Waste as a Component of Growing Media: Manifestations, Assessments, and Sources of Their Phytotoxicity"

_plants, 2024, doi:10.3390/plants13142000_

Round 1

Reviewer 1 Report

Comments and Suggestions for Authors

 The manuscript of Juncheng Liu et al represents the literature review, illuminating number of issues, dedicated to the possibilities of using plant-derived wastes as a component of growing media, with the emphasis of their possible phytotoxicity. The topic is relevant and doubtless is of interest for the readership of Plants. However, in its current state, the paper is too general and superficial and should be revised.

In Introduction and partly in section 4 authors provide comprehensive information about the global amount of plant-derived wastes, reasons and advantages of their use as components of growing media and potential concerns due to the manifestation of  phytotoxicity. But further, coming directly to the phytotoxicity issues, authors do not provide enough specific data, instead giving well-known   and sometimes even confusing information. Namely:

 Lines 89-91: “germination inhibition manifests … leaf yellowing and browning, root curling, and root shortening. Conversely, growth inhibition typically presents as leaf discoloration and stunted plant stature”: these effects occur not only due to presence of toxic substances, but also due to the deficit or improper balance of nutrients.

L. 103: “The manifestation of phytotoxicity is contingent upon the form and concentration of the source substance”  - well-known statement, concretization is needed. Which particular forms and concentrations, for which group of toxicants? Obviously, they would be different, for example, for heavy metals and organic toxicants. Among the latter can be dangerous contaminants like PAH or pesticide residues, as well as hormone-like or other biologically active substances, stimulating in low concentrations and toxic in high.

Section 3: Methods of Phytotoxicity Assessment is also superficial. L 138-140: “samples exhibiting increased concentrations of organic acids and amino acids were consistently correlated with heightened phytotoxicity levels”. Which particular organic acids and amino acids, in which concentrations, please? Amino acids are  usually considered as an N-source and not toxic.

The description of bioassays (Seed germination and Seedling growth experiments) should be supplied with a table, summarizing numerous existing protocols. Authors mention several ISO and other corresponding guidelines,  but their specific focus, applicability and other important details is not possible to understand well from the plain text.  Which test-cultures are recommended, which test-responses? What are advantages, disadvantages and applicability of direct contact, sandwich and aqueous extract methods?

L. 273-274: “phenolic compounds and organic acids have been identified as principal phytotoxic agents” – again, too general. Which particular compounds, high- or low-molecular, in which concentrations?

Please clarify, why “ elevated electrical conductivity (EC) levels are the primary factor contributing to the phytotoxicity observed in mushroom residues” (l/354-355). Elevated EC can be caused not only by toxic salts, but also by nutrients (salts of potassium, ammonia, nitrates).

Finally, conclusions are also too general and do not specify the novelty of paper and findings of authors. This section should be rewritten.

Author Response

Comments 1: In Introduction and partly in section 4 authors provide comprehensive information about the global amount of plant-derived wastes, reasons and advantages of their use as components of growing media and potential concerns due to the manifestation of  phytotoxicity. But further, coming directly to the phytotoxicity issues, authors do not provide enough specific data, instead giving well-known   and sometimes even confusing information. Namely: Lines 89-91: “germination inhibition manifests … leaf yellowing and browning, root curling, and root shortening. Conversely, growth inhibition typically presents as leaf discoloration and stunted plant stature”: these effects occur not only due to presence of toxic substances, but also due to the deficit or improper balance of nutrients.

Response 1:Thank you for pointing this out. l agree withthis comment. Therefore, to achieve this objective, I have incorporated pivotal new data, specifically within the ranges of lines 75 to 80 and 96 to 99 in the revised manuscript

Comments 2: L. 103: “The manifestation of phytotoxicity is contingent upon the form and concentration of the source substance”  - well-known statement, concretization is needed. Which particular forms and concentrations, for which group of toxicants? Obviously, they would be different, for example, for heavy metals and organic toxicants. Among the latter can be dangerous contaminants like PAH or pesticide residues, as well as hormone-like or other biologically active substances, stimulating in low concentrations and toxic in high.

  1. 273-274: “phenolic compounds and organic acids have been identified as principal phytotoxic agents” – again, too general. Which particular compounds, high- or low-molecular, in which concentrations?

Response 2:I appreciate your proposed amendment, and I consider it with utmost seriousness. In response, I have incorporated data from diverse researchers regarding the adverse effects of specific organic acids and phenolic compounds on plants. This addition aims to offer readers a comprehensive and intuitive understanding of how these compounds may react negatively with plants at specific concentration thresholds. These revisions are meticulously presented in Table 3 within the text.

Comments 3: Section 3: Methods of Phytotoxicity Assessment is also superficial. L 138-140: “samples exhibiting increased concentrations of organic acids and amino acids were consistently correlated with heightened phytotoxicity levels”. Which particular organic acids and amino acids, in which concentrations, please? Amino acids are  usually considered as an N-source and not toxic.

Response 3: I wholeheartedly support this proposed amendment. In response, I have meticulously revised the paragraph, explicitly identifying the names and concentrations of specific organic acids. These modifications are now reflected in the revised lines 143 to 148.

Comments 4: The description of bioassays (Seed germination and Seedling growth experiments) should be supplied with a table, summarizing numerous existing protocols. Authors mention several ISO and other corresponding guidelines,  but their specific focus, applicability and other important details is not possible to understand well from the plain text.  Which test-cultures are recommended, which test-responses? What are advantages, disadvantages and applicability of direct contact, sandwich and aqueous extract methods?

Response 4:I appreciate your proposed amendment. In response, I have compiled a table delineating the strengths and weaknesses of various assessment methods. This table aims to facilitate readers’ understanding of the distinct approaches employed in our research(table 2)

Comments 5: Please clarify, why “ elevated electrical conductivity (EC) levels are the primary factor contributing to the phytotoxicity observed in mushroom residues” (l/354-355). Elevated EC can be caused not only by toxic salts, but also by nutrients (salts of potassium, ammonia, nitrates).

Response 5: Thank you for promptly identifying the error in the article. I have made the necessary revisions. Mushroom residues often exhibit electrical conductivity (EC) levels that surpass the optimal growth range for plants, thereby potentially constraining plant development. The phytotoxic effects associated with elevated EC are indirectly attributed to an accumulation of inorganic salts.

Comments 6: Finally, conclusions are also too general and do not specify the novelty of paper and findings of authors. This section should be rewritten.

Response 6: Thank you for your critique and suggestions. In response, I have revised the concluding chapter to explicitly highlight the significance and outcomes of this study. Furthermore, I have provided recommendations for future research directions in this field.

Reviewer 2 Report

Comments and Suggestions for Authors

Plant-Derived Waste as a Component of Growing Media:  Manifestations, Assessment, and Sources of Their Phytotoxicity

Authors mention straw (mainly rice and wheat straw) garden wastes and spent mushroom substrates as sources of phytotoxic compounds released during decomposition. Whereas maize and wheat straw may remain on the fields (then at low concentration with little or no phytotoxic effect – general method of modern agriculture if straw is not used otherwise) garden wastes and mushroom spent substrates are generated where the food is harvested for distribution at locally concentrated amounts with a high phytotoxicity. Authors describe internationally authorized test assays and tow main categories of toxic compounds (phenolic compounds and acetic acids) but even in their conclusions and prospects they do not mention how to deal with phytotoxicity – or how world-wide is dealt with this problem. They do not mention composting or anaerobic digestion as a treatment process to minimize plant toxicity. After a proper treatment the statement “Efficient utilization of plant waste necessitates precise analysis and evaluation of its phytotoxic properties” would be appropriate…….Such analysis is vital to negate their phytotoxic effects and to lay a quantitative groundwork for the application of these wastes as growth media”.
I agree with most of what authors stated but I think it is known world-wide that fresh plant material may or is plant-toxic and thus toxicity test would thus be logic and economically useful after treatment of toxicity reduction.

Author Response

Thank you for bringing this to my attention. I concur with the comment, and as a result, I have made appropriate revisions. In the revised section, I have provided specific data on the effects of straw and mushroom residue on plants. Additionally, I have included a tabulated summary in the article, outlining the impacts of phenolic compounds and organic acids on plant growth based on existing research. Furthermore, I have reworked the conclusion chapter, explicitly stating that composting is currently the most widely used method for mitigating plant toxicity.

Reviewer 3 Report

Comments and Suggestions for Authors

The review titled “Plant-Derived Waste as a Component of Growing Media: Manifestations, Assessment, and Sources of Their Phytotoxicity”, is an interesting and well written paper. This review summarizes research insights that contribute to a better understanding of phytotoxicity and its consequences and provides valuable references and guidelines for future research and practical application. This study describes various evaluation methods and procedures for the evaluation of plant waste phytotoxic properties. The findings of the study contribute to the advancement of a circular economy. Truly assessed plant waste as a renewable organic resource can be included in the composition of the growing media without negative impact to plant safety.

This review can be published at it’s present form. The only remark is that this paper doesn’t contain the yield quality (fruit, grain) assessments. To fulfil the description of the plant phytotoxicity evaluations, I suggest including plant productivity and yield quality methods in “Methods of Phytotoxicity Assessment”.

Author Response

Thank you for your endorsement and advice on this article. I agree with you, and for this reason I have revised this paper by adding table2, in which I summarise existing plant assessment methods.

Round 2

Reviewer 2 Report

Comments and Suggestions for Authors

New citations are not correctly formatted:

Line 561,562:Journal number and pages missing

Line 601-603: year of publication missing

Line 616, 617: Omitt words in capital letters, arrangement of authors, title, journal and journal number/pages wrong.

Comments on the Quality of English Language

in my opinion it can be published

Author Response

Thank you very much for your advice, I agree with this and I have made the amendments as suggested